# The Primary Healthcare Physician’s Awareness and Engagement in Community-Based Medical Education: A Mixed Qualitative and Quantitative Study

**DOI:** 10.3390/healthcare11192676

**Published:** 2023-10-03

**Authors:** Elhadi Miskeen, Abdullah M. Al-Shahrani

**Affiliations:** 1Department of Obstetrics and Gynecology, College of Medicine, University of Bisha, Bisha 67714, Saudi Arabia; 2Department of Obstetrics and Gynecology, Faculty of Medicine, University of Gezira, Wad Medani 21111, Sudan; 3Department of Family Medicine, College of Medicine, University of Bisha, Bisha 67714, Saudi Arabia; ab_alshahrani@ub.edu.sa

**Keywords:** community-based medical education, primary healthcare physicians, Bisha, Saudi Arabia

## Abstract

Background: Community-based medical education (CBME) is an essential component of medical education, where primary healthcare physicians (PCPs) play a crucial role. This study explores PCPs’ awareness and engagement in CBME and investigates the factors influencing their participation. Methods: This mixed study was conducted in two phases. In the first phase, a qualitative study was conducted using semi-structured interviews with PCPs as well as thematic analysis related to their awareness and engagement in CBME. In the second phase, a quantitative survey was conducted on 72 PCPs’ pre- and post-training programs. Results: Primary healthcare physicians had a positive attitude toward community-based medical education. The participants had an acceptable level of awareness and engagement, which increased substantially by (*p*-value = 0.03) and (*p*-value = 0.003), respectively. Logistical analysis indicated that non-Saudi physicians were more likely to participate in the CBME (*p*-value = 0.001). Professions and academic experiences influenced their willingness to engage and participate in CBME (OR = 7.5, *p*-value = 0.001) and (OR = 0.21, *p*-value = 0.001), respectively. Conclusion: The study findings highlighted the importance of increased awareness and the factors that enhance PCPs’ engagement in CBME. This positive perspective of the PCPs will help build effective partnerships and facilitate the extension of the curriculum to apply CBME.

## 1. Introduction

Community-based medical education (CBME) is a type of medical education that focuses on providing students with hands-on experience in the community. This type of education emphasizes learning from and engaging with patients, families, and other community members. It also encourages students to understand the social determinants of health such as poverty, racism, and access to healthcare. Community-based medical education can occur in various settings including hospitals, clinics, schools, and community centers [1]. CBME is an approach that emphasizes the importance of involving local communities in the educational process [2]. It is a collaborative effort between schools, families, and community members to create a learning environment that is relevant, meaningful, and responsive to the community’s needs [3].

CBME recognizes that education is about academic achievement and developing social skills, cultural awareness, and civic responsibility [4]. CBME aims to empower individuals and communities to actively shape their educational experiences and outcomes [5]. This approach effectively improves student engagement, academic performance, and well-being [6].

PHPs are critical in community-based medical education, focusing on curriculum design to promote social accountability in health profession education. This becomes especially crucial in terms of promoting cultural diversity, emphasizing the importance of their involvement in CBME initiatives [7]. They are the first point of contact for patients seeking medical care and are responsible for providing comprehensive and continuous care to their patients. As such, they have a unique perspective on the health needs of their community and can provide valuable insights into the development of community-based medical education programs [8].

One of the key roles of primary healthcare physicians in community-based medical education is to serve as mentors and preceptors for medical students and residents [9]. They can guide and support these learners as they develop their clinical skills and knowledge. This includes teaching them how to conduct patient assessments, diagnose illnesses, develop treatment plans, and communicate effectively with patients [9].

In addition to serving as mentors, primary healthcare physicians can contribute to developing community-based medical education programs by sharing their knowledge and expertise with educators [10]. They can provide insights into the health needs of their community, identify areas where additional training may be needed, and help educators design relevant and effective programs [11].

Primary healthcare physicians can be vital in promoting community engagement among medical learners [12]. By involving learners in community outreach activities such as health fairs or vaccination clinics, they can help them to develop a deeper understanding of the social determinants of health, and the importance of working collaboratively with other healthcare providers [13].

Primary healthcare physicians are essential in community-based medical education [14]. With their mentorship, expertise and engagement with learners and educators alike, they can help to ensure that future generations of healthcare providers are well prepared to meet the needs of their communities [15].

The primary health care system in Saudi Arabia is an established and effective system that provides opportunities for PCPs to participate in CBME activities with greater integration into the medical school. CBME can build partnerships between the university, service providers, and community, whilst the students’ learning and service activities can positively influence community health and prepare them to care for people in rural communities. Saudi Vision 2030, universal health coverage, and primary healthcare centers are interconnected and synergistic components of the country’s healthcare transformation. By prioritizing UHC and investing in PHCs, Saudi Arabia aims to provide accessible and equitable healthcare services to its citizens, contributing to a healthier, more resilient, and prosperous society. In this context, this study explored the awareness and engagement of PCPs toward community-based medical education and investigated the factors influencing their participation in the CBME organized by the College of Medicine, University of Bisha.

## 2. Materials and Methods

Study design: This mixed study was conducted in two phases from June 2022 to December 2022. In the first phase, a qualitative study was conducted using semi-structured interviews with PCPs as well as thematic analysis related to their awareness and engagement in CBME. In the second phase, a quantitative survey was conducted on the participants’ pre- and post-training programs.

Participants and sampling: All physicians working in primary healthcare centers (PC) in Bisha province were invited to participate in the study conducted by the University of Bisha (n = 72) to identify the predictors of primary healthcare physicians for engagement in community-based medical education. They work in 33 PCs with clinical or clinical duties and administrative tasks. They provide general practice, family medicine, obstetrics, and services.

The sample size was determined using a power analysis based on the estimated population size of primary healthcare physicians working in community health centers. A random sampling technique was used to select participants from different PCs. The sample size was determined according to the following points: the expected difference in awareness and engagement scores before and after the training programs, the desired significance level set at 0.05, and the desired power of the study set at 0.80. The sampling procedure involved identifying and recruiting primary healthcare physicians (PHPs) from relevant PHCs. The participants were selected using a systematic or random sampling method to ensure representative and unbiased inclusion in the study. PHPs who met the eligibility criteria were invited to participate voluntarily in the study.

Training workshops: We designed a program of three serial workshops concerning community-based medical education. An assessment was performed at the beginning of the first workshop and at the end of the third workshop. In between, a hotline was active to respond to their questions.

Key points from the workshop emphasized the importance of community-based medical education for training primary healthcare physicians to engage with their communities and commit to improving patient outcomes. They were asked to complete the questionnaire containing the same items both pre- and post-evaluation. The main areas covered in the training program included:Introduction to CBME:Understanding the community:Engaging with the community:Curriculum development:Faculty development: how to train faculty members to teach in a CBME program.Evaluation and assessment: how to evaluate and assess the effectiveness of a CBME.Sustainability focused on how to ensure that the program is viable over time.

Data collection: The quantitative data collection utilized a self-administered questionnaire. The questionnaire was distributed to participants through their workplace or via email. Participants had the option to complete the questionnaire online or on paper.

The questionnaire included the following sections:The sociodemographic data and general characteristics of the study population, including: age, nationality, professional position, gender, years of experience, place of work, and academic experienceAwareness towards CBME, including: the influence of CBME on the student selection for rural placement, CBME as an additional opportunity for students going on a rural placement, and increasing the interest of graduates to work in a rural areaPCP intention to engage in the CBME program at the University of Bisha and willingness to participate in training medical students at the University of Bisha.

Qualitative data were collected by using open question surveys before the training programs in order to provide a comprehensive understanding of how community-based medical education programs impacted primary healthcare physicians’ awareness and engagement levels. In each PC, the number of physicians varied from 2 to 4. We recruited participants for the quantitative survey by inviting all primary healthcare physicians from the selected PC to engage. The total number of PHPs participating in the quantitative study was 25 based on specific criteria related to their experiences or roles in CBME.

Data Analysis: The quantitative data were analyzed using Stata/BE 17.0 Stata Corp LLC., College Station, TX, USA. Descriptive statistics were summarized. Logistic regression analysis identified predictors of primary healthcare physicians’ engagement in community-based medical education.

Via a thematic analysis approach, the qualitative analysis identified three main themes related to PCPs’ awareness and engagement in CBME: (1) lack of knowledge about CBME; (2) limited involvement in CBME activities; and (3) perceived benefits of engaging in CBME.

## 3. Results

In this study, 72 PCPs were enrolled and participated in a comprehensive educational program. More than half were under 40 (54.2%), and most were non-Saudi (91.7%). Most were female (68.1%) and worked in urban settings (66.7%). Regarding their work experience, we found that more than half had 10 years of experience (63.9%). In contrast, most (88.9%) performed purely clinical duties, and 83.3% had no academic experience (Table 1).

The training program influenced PCPs’ willingness to participate in community-based medical education activities at the College of Medicine, University of Bisha. The proportion of PCPs who desired to engage in CBME increased from 83.3% to 94.4% (*p*-value = 0.001). An age of 40 (*p*-value = 0.05), female gender (*p*-value = 0.002), pure clinical duties (*p*-value = 0.001), working in urban settings (*p*-value = 0.001), and academic experience (*p*-value = 0.001) were significant predictors of willingness. However, nationality and years of experience did not demonstrate substantial correlations (*p*-values greater than 0.05), although willingness to participate was positively reflected and increased after the training program (Table 2). The training program’s impact on the awareness and desire to engage PCPs in the CBME is explained in Figure 1.

Considering the importance of awareness in CBME engagement, most PCPs had an acceptable level of understanding, which increased substantially (*p*-value = 0.03) from 66.6% to 81.9% after the training. The majority (66.6% vs. 87.5%) of PCPs reported that students’ community involvement influenced their decision in choosing a rural placement after training (*p*-value = 0.001). Most participants were aware that the CBME encouraged students to train in PC and rural institutions (*p*-value 0.001). In addition, most participants knew that CBME offers pre- and post-training opportunities for students and graduates to work in rural placements (86.1%, 97.2%, and 75%, 93%). (Table 3).

Logistic regression models were used to identify factors that positively predicted the PCPs willing to engage in CBME activities at the University of Bisha. Binary logistic analysis revealed that non-Saudi physicians were 10.33 times more likely to be willing to be a part of CBME than Saudi physicians (*p*-value = 0.001, 95% CI = 4.47–23.88). However, the willingness to participate is significantly higher among physicians with work experience of more-than 10 years (OR = 1.61, *p*-value = 0.05, 95% CI = 0.99–2.63). Professions with pure clinical duties versus clinical and administrative showed a highly significant willingness to participate in CBME (OR = 7.5), (*p*-value = 0.001, 95% CI = 3.586–15.68). PCPs with academic experience were more willing to participate in CBME than those with non-academic experience (OR = 0.21, *p*-value = 0.001, 95% CI = 0.11–0.39). Age and place of work were not strong predictors for detection of the willingness to participate in CBME (OR = 0.94, *p*-value = 0.08, 95% CI = 0.58–1.51), (OR = 0.14, *p*-value = 0.1, 95% CI = 0.01–1.5), respectively. Further details of logistic regression models can be seen in Table 4.

The qualitative analysis identified that PCPs had a positive attitude toward CBME. They believed it was an effective way to improve the quality of healthcare services in the community and to enhance their professional development. The physicians also expressed a willingness to participate in community-based medical education activities, such as teaching and mentoring medical students and residents. The predictors of primary healthcare physicians’ engagement in community-based medical education included their experience level, workload, the availability of resources, and support from their organizations. Physicians with more experience were more likely to engage in community-based medical education activities, whereas those with heavier workloads were less likely to do so. Support from their organizations, including recognition and incentives for participation, was crucial for motivating physicians to engage in community-based medical education. However, further efforts are needed to address the barriers that prevent physicians from engaging fully in these activities such as workload and resource constraints. Organizations should also provide support and incentives for physicians who engage in CBME to ensure its sustainability over time (Table 5).

## 4. Discussion

In Saudi Arabia, primary health care physicians play a crucial role in the healthcare system by providing direct care services to individuals and families. They are often the first point of contact for patients seeking medical attention. They are responsible for diagnosing and treating common illnesses, managing chronic conditions, and promoting preventive health measures. CBME approach aims to prepare medical students for the challenges they will face in their future careers by providing them with hands-on experience working with diverse patient populations and addressing the social determinants of health. In this discussion, we will explore the awareness of, engagement with, and the influencing factors for PCPs toward CBME. Further in-depth analysis will consider the qualitative factors of CBME in a newly established medical school in Saudi Arabia.

In this study, most PCPs were adequately aware of CBME, and this only increased following training. This finding is inconsistent with the results of (Bansal A et al., 2022), indicating that the activity could significantly raise primary healthcare physicians’ awareness of community-based medical education [16]. (Reis et al., 2022) revealed that the movement can play a crucial role in increasing the awareness of primary healthcare physicians toward community-based medical education [17]. Our training program provides the participants with the necessary skills and knowledge, aiming to improve patient healthcare outcomes and promote more effective collaboration between providers and communities. This is allied with the vision of the University of Bisha as a medical college established in the rural service areas of Saudi Arabia.

In this study, the participants reported that the involvement of students in their communities influenced their decision to choose a rural placement, and that community-based medical education can provide opportunities for students and graduates to work in rural stations. This finding is consistent with a recent study from the USA which indicated that the involvement of students in their communities could significantly influence their decision in choosing a rural placement [18]. Also, another study from Australia showed that community-based medical education programs provided opportunities for students and graduates to work in rural stations [19]. When students engage with the local community, they gain a deeper understanding of a rural population’s healthcare needs and challenges. This exposure can help them to develop a sense of purpose and a commitment to assisting these underserved communities. Our programs offer hands-on training and experience in rural healthcare settings that can help students to develop the skills and knowledge needed to provide quality care in these areas. Furthermore, community-based medical education programs often offer mentorship and support for students as they navigate the challenges of working in rural settings.

This study demonstrated that the training program increased the willingness of PCPs to participate in community-based medical education activities conducted by the College of Medicine at the University of Bisha. These findings, consistent with the study, indicated that the engagement of primary healthcare physicians in community-based medical education was crucial for developing competent and compassionate healthcare professionals [20]. Study for assessment of health workers’ perspective on their motivation in community-based primary health care in rural Malawi, showed that engaging primary healthcare providers in community-based medical education also benefits the providers themselves [21]. The impact of training to increase the engagement of primary healthcare physicians in community-based medical education was confirmed [22,23,24]. The concentration of primary healthcare physicians in community-based medical education is essential for building a robust healthcare workforce equipped to meet the diverse needs of the population. Medical schools and primary healthcare providers must collaborate to create meaningful learning experiences that benefit both students and patients.

Significant predictors of willingness to engage were an age over 40, female gender, pure clinical duties, working in urban settings, and academic experience. However, nationality and years of experience did not have significant correlations. These findings have been highlighted in the literature, and several important factors can influence primary healthcare physicians’ willingness to participate in community-based medical education [24,25]. The College of Medicine, University of Bisha, will consider these factors and work towards addressing them to encourage more primary healthcare physicians to participate in community-based medical education programs.

An in-depth qualitative analysis highlighted the factors that impacted the interest and motivation of primary healthcare physicians towards engagement in CBME activities at the University of Bisha. It indicated that physicians’ interest benefitted their professional development, patient care, effective community engagement, receiving institutional support and having enough spare time to engage in CBME.

This study consisted of findings that revealed several factors that impacted the interest of primary health physicians in community-based education [26,27]. This qualitative approach involved in-depth interviews with primary health physicians with community-based education experience. We can use these findings to better understand the environment that can help with the adoption of CBME at the University of Bisha. Physicians who participate in community-based medical education programs benefit from the experience and have the opportunity to mentor and teach the next generation of healthcare providers, which can be a rewarding experience. They may also gain new insights into the healthcare needs of their community and can develop new strategies for addressing those needs.

This study indicated that physicians who are firmly committed to the local community are more likely to participate in community-based medical education programs. The relationship between physicians and a strong commitment to their local community has been extensively evaluated in the literature [28,29,30,31]. It suggests that they are more likely to participate in community-based medical education programs because they see it as an opportunity to positively impact the health of their neighbors and build stronger connections within their professional network. The CBME programs at the University of Bisha provide opportunities for medical students and residents to gain hands-on experience in community settings such as clinics, hospitals, and community health centers. By working alongside experienced physicians in these settings, students can learn about the local population’s unique health needs and challenges as well as developing critical clinical skills.

The College of Medicine at the University of Bisha highlighted that engaging students in community-based medical education and offering opportunities to work in local communities can be a potential solution for tackling the shortage of healthcare professionals in rural areas. By fostering a sense of connection and commitment to these communities, we can encourage more students to choose rural placements and ultimately improve access to healthcare for underserved populations. The curriculum of the College of Medicine, University of Bisha, will be updated according to the community-based medical education curriculum. The current situation at the University of Bisha is suitable for students to train in their community settings. The findings of this research will help us to update the curriculum at the College of Medicine, Saudi Arabia. The curriculum update focuses on at least one of the components of community-based medical education such as primary healthcare centers, families, and rural hospitals.

## 5. Conclusions

The study findings highlighted the importance of increased awareness and the factors that enhanced PCPs’ engagement with CBME. This positive perspective of the PCPs helps build effective partnerships and facilitates the extension of the curriculum in applying CBME. It also emphasizes the need to address the barriers that prevent PCPs from engaging in CBME activities. Future research should focus on developing strategies to increase PCPs’ engagement in CBME and evaluating the impact of such strategies on medical education and healthcare outcomes.

## Figures and Tables

**Figure 1 healthcare-11-02676-f001:**
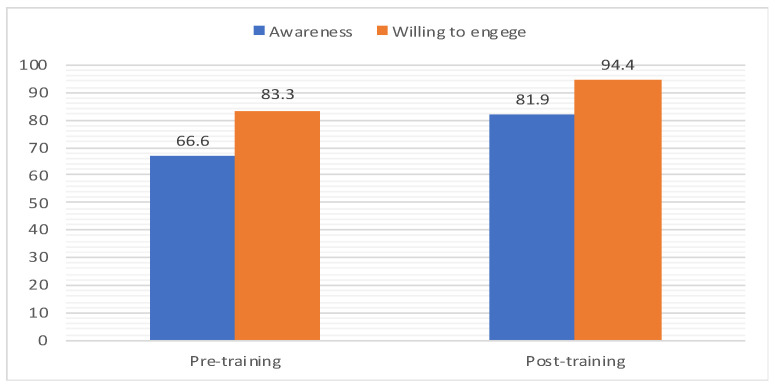
The impact of the training program on the awareness and willingness to engage PCP in the CBME (n = 72).

**Table 1 healthcare-11-02676-t001:** Characteristics of the participants.

Variables		No.	%
Age	<40 years	33	45.8
	≥40 years	39	54.2
Nationality	Saudi	6	8.3
	Non-Saudi	66	91.7
Gender	Female	49	68.1
	Male	23	31.9
Work experience	≥10 years	46	63.9
	<10 years	26	36.1
Professional	Pure clinical duties	64	88.9
	Clinical & administration	8	11.1
Place of work	Urban	48	66.7
	Rural	24	33.3
Academic experience	Yes	12	16.7
	No	60	83.3
Total		72	100

**Table 2 healthcare-11-02676-t002:** The relationship between the participant characters and their willingness to participate in CBME (n = 72).

General Characteristics		Willingness to Participate in CBME Activities				
		Pre-		Post		*p*-Value
		Yes No. (%)	No No. (%)	Yes No. (%)	No No. (%)	
Age	<40 years	31 (43)	2 (2.7)	33 (45.8)	0 (0)	0.05
	≥40 years	29 (40.3)	10 (14)	35 (48.5)	4 (5.6)	
Nationality	Saudi	4 (5.5)	2 (2.7)	6 (8.3)	0 (0)	0.64
	Non-Saudi	56 (77.8)	10 (14)	62 (86.1)	4 (5.6)	
Gender	Female	43 (59.7)	6 (8.35)	48 (66.6)	1 (1.4)	0.002
	Male	17 (23.6)	6 (8.35)	20 (27.7)	3 (4.2)	
Experience	≥10 years	37(51.4)	9 (12.5)	43 (59.8)	3 (4.2)	0.82
	<10 years	23 (31.9)	3 (4.2)	25 (34.6)	1 (1.4)	
Professional	Pure clinical	54 (75)	10 (14)	62 (86.1)	2 (2.8)	0.001
	Clinical & administration	6 (8.3)	2 (2.7)	6 (8.3)	2 (2.8)	
Place of work	Urban	46 (63.8)	2 (2.7)	47 (65.3)	1 (1.4)	0.001
	Rural	14 (19.5)	10 (14)	21 (29.1)	3 (4.2)	
Academic experience	Yes	10 (13.9)	2 (2.7)	12 (16.7)	0 (0)	0.001
	No	50 (69.4)	10 (14)	56 (77.7)	4 (5.6)	
Total		60 (83.3)	12 (16.7)	68 (94.4)	4 (5.6)	0.001

**Table 3 healthcare-11-02676-t003:** The awareness of PCPs about CBME (n = 72).

Items	Participant Awareness			*p*-Value
	Response	Pre- No. (%)	Post- No. (%)	
Level of awareness regarding CBME curriculum	Fully aware	48 (66.6)	59 (81.9)	0.03
	Not aware	24 (33.4)	13 (18.1)	
Community involvement of student has an impact on students to choose a rural placement	Yes	48 (66.6)	63 (87.5)	0.001
	No	14 (19.4)	9 (12.5)	
CBME encourage the student to train in the PC and Rural hospital	Yes	64 (88.9)	70 (97.2)	0.001
	No	8 (11.1)	2 (2.8)	
CBME provide opportunities for students to work in rural placement	Yes	62 (86.1)	70 (97.2)	0.001
	No	10 (13.9)	2 (2.8)	
CBME provide opportunities for graduates to work in rural placement	Yes	54 (75)	67 (93)	0.001
	No	18 (25)	5 (7)	
Total		72 (100)	72 (100)	

**Table 4 healthcare-11-02676-t004:** Logistic regression for the PCPs physicians willing to engage in CBME activities.

Variables	Odd Ratio	*p*-Value	[95% Confidence Interval]	
			Upper	Lower
Age, ≥40 years versus <40 years	0.943	0.808	0.5860165	1.516987
Nationality, Non-Saudi versus Saudi	10.33	0.001	4.470051	23.88737
Gender, female versus male	7.2	0.09	0.7057967	73.44891
Work experience, ≥10 years versus <10 years	1.61	0.05	0.9905255	2.634427
Professions, pure clinical versus clinical and administration	7.5	0.001	3.586581	15.68346
Place of work, rural versus urban	0.148	0.108	0.0146233	1.5169
Academic experiences versus non-academic experience	0.214	0.001	0.1148741	0.3997277

**Table 5 healthcare-11-02676-t005:** Qualitative result about predictors of PCPs toward engagement in CBME.

Theme	Code	Result
Personal interest and motivation	Motivation	Physicians who have a personal interest in teaching and learning are more likely to engage in community-based medical education.
Institutional support	Received	Physicians who receive institutional support from their healthcare organization or academic institution are more likely to engage in community-based medical education.
Time availability	Sufficient time	Physicians who have sufficient time available to engage in community-based medical education are more likely to participate.
Professional development opportunities	Opportunity for CPD	Physicians who perceive that community-based medical education will benefit their professional development, patient care, and effective community engagement are more likely to participate.
Personal characteristics:	Communication skills, leadership skills positive attitude	Physicians with good communication skills, leadership qualities, and a positive attitude towards teaching and learning are more likely to engage in community-based medical education.
Community engagement	Engagement	Physicians who have a strong engagement with the local community and a desire to give back are more likely to participate in community-based medical education programs.

## Data Availability

Not applicable.

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
