# Peer review of "The Primary Healthcare Physician’s Awareness and Engagement in Community-Based Medical Education: A Mixed Qualitative and Quantitative Study"

_healthcare, 2023, doi:10.3390/healthcare11192676_

Round 1
Reviewer 1 Report
Congratulations for your interesting article.
I owuld like to suggest for line 86-94. Participants and sampling.
Usually researcher calculate the sample size based on power, confidence limit, population , margin of erra we put into a soft ware. Thus you should mention which sample size calculation software or web site you use, what data did you feed into it and the reason for it. (For example you key in 100 in population as your estimated Primary helath care physician in that area is around 100.) When calculated sample size is X which is less than actual population 100 you may need to use sampling procedure. You mentioned that you use "random sampling". But you need to be more specific is it systematic random sampling or simple random sampling? You need to metion the detail procedure of your systematic random sampling or simple ranom sampling so that reader will be able to do the similar research. The sequence or your writing should be sample size calculation followed by sampling procedure.
You mentioned that your participants worked in 33 PHC centres. How many primary care physicians in each clinics?. Did you take one randomly pick physician from each clinic? or did you randomly selected PHC and invited allfrom randomly selected PHC? Your study is mixed method so you need to mentioned how did you select participants for qualitative study and how many participants participated in it, how many prtiicpated in quantative part.
Line 196 table 2, I wonder why do you want to group into less than 40 years and 40 or above? Alos work experiences why do you want to group into less than 10 years and 10 years and above? Age and year of work experiences are linked. ( I am not quite sure about Saudi but in my context to become primary care physician specialist they need at least 8 yrs to 10 yrs after medical degree. Age of studnets at gradaution is 26 or 28. The older the person may be the most experiences person ) If you want to see the actual influence of age you may need to stratified work experiences.
Author Response
We sincerely appreciate the valuable comments provided by the reviewers, which have significantly enriched the content of the paper.
Best.

Reviewer 2 Report
Throughout the manuscript the authors refer to "primary healthcare physicians" and "PHC", when they are actually describing "primary care physicians" and "primary care". PHC is an approach or a concept (similar to universal health coverage. Therefore there is no such thing as a "PHC provider" or a "PHC facility". Please substitute "PC" for "PHC" physician.
The other issue is the references to the University of Bisha. While it is important to mention this once, it does not need to be repeated. The sentence in lines 255 - 258 could simply be written "The College of Medicine, University of Bisha will consider....."
Also, first person references should be rephrased, for example, the sentence starting in line 284 should not being with "We argued...."
The paragraph starting in line 48 is redundant with the paragraph starting in line 68. The citations ([7] and [14])can all be used in the first occurrence.
The first occurrence of PHP (in line 73) should be spelled out and it should be PCP (primary care physician).
There are multiple instances where the tense of the verb is the present tense when it should be in the past. (for example in line 110, line 111, line 113, line 114)
Author Response
Dear reviewer, we extend our sincere gratitude for your valuable comments, which have greatly enhanced the content of our paper. We have addressed all of your comments in our response.
Best
